

# Total Reflection of the Strahl within the Foot of the Earth's Bow Shock

Christopher A. Gurgiolo[*1], Melvyn L. Goldstein[2], and Adolfo Viñas[3]

[1]Bitterroot Basic Research, Hamilton, MT, USA

[2]Space Science Institute, Boulder, CO, USA

[3]Department of Physics, American University, Washington, DC 20016, USA, and Geospace
Physics Laboratory, NASA Goddard Space Flight Center, Greenbelt, MD, USA

*Correspondence to:* Melvyn L. Goldstein
(u2mlg1@gmail.com)

[*]Deceased

**Abstract.** The reflection of a fraction of the solar wind at the bow shock to some extent defines the
physical properties of what is known as the foreshock, the region where the interplanetary magnetic
field has a direct connection to the bow shock. Both ion and electron reflection have been observed
and together form a significant source of free energy that is responsible for many of the instabilities

observed in this region. In this paper we concentrate on reflection of electrons at the shock and report
two significant findings: The first is that the strahl, the field aligned component of the electron solar
wind distribution, is fully reflected at the bow shock; the second finding is that the reflection occurs
in the foot of the shock and not in the shock ramp. The latter implies that mirroring appears to play,
at most, only a minor role in the electron reflection process.

**1  Introduction**

The region upstream of a planetary bow shock that is magnetically connected to the shock is known
as the foreshock (Russell et al., 1971). This is a highly dynamic and often turbulent region charac-
terized not only by a solar wind presence but also by the presence of ion and electron distributions
formed from solar wind particles reflecting at the shock and propagating back into the upstream solar

wind along the magnetic field (Paschmann et al., 1981; Thomsen et al., 1983; Gedalin, 2016). These
latter distributions may also include particles that have leaked back upstream from downstream of the
shock (Gosling et al., 1989). We will not distinguish these distributions, and will, for convenience,
label them simply as *return* distributions.

    Although partial reflection of both solar wind ions and electrons off Earth's bow shock has been





recognized for many years (Gosling et al., 1978; Bonifazi and Moreno, 1981a,b; Anderson et al., 1985; Gosling et al., 1989) most studies have focused on ion observations because:

1. The solar wind ion velocity distribution function (VDF) is generally less complex than that of the electrons, despite the fact that the ion distributions do vary with solar activity when fast streams may contain field aligned beams and heavier ions, such as Helium. On the other hand, the electron VDF are generally far from Maxwellian in shape and typically consists of three component populations; core, halo, and strahl that together can extend from a few eV into the keV range.

2. The ions have substantially longer gyro-periods, which allows for detailed studies of the evolution of the post reflection VDFs by typical electrostatic analyzers that often require seconds to acquire a full 3D VDF.

3. The solar wind ions, being of much higher energy than the electrons, are less susceptible to issues that often plague measurements of the solar wind electrons, such as spacecraft charging, low energy photo-electrons, noise, etc.

It has generally been assumed that results obtained from the ion studies are also applicable to the reflection of electrons from the shock, but this assumption has not been intensively investigated.

At least for ions, the specifics of the reflection of the solar wind at the bow shock are dependent, to some degree, on whether the reflection occurs in a quasi-parallel or quasi-perpendicular shock configuration (see eg., Fuselier and Schmidt, 1994). In specular reflection off a quasi-perpendicular portion of a shock, the guiding center of the reflected particle is directed downstream often allowing for multiple reflections off the shock and the reflection efficiency can reach as high as 20% of the incident solar wind (Paschmann and Sckopke, 1983). Reflections off a quasi-parallel shock show a much lower reflection efficiency and result in a guiding center motion which is predominantly directed upstream (Gosling et al., 1982; Schwartz and Marsch, 1983). Yuan et al. (2007), using simulations have shown electron reflection percentages as high as 10%, the percentage being dependent on the size of the shock magnetic field overshoot.

Return distributions are almost always field-aligned and are often non-gyrotropic for both electrons and ions. The anisotropy basically is the result of gyrophase-bunching that can occur in the reflection process (Gurgiolo et al., 1983) and is integral to the formation of the partial and full ring distributions observed in the foreshock. Those distributions are thought to be derived from the phase-mixing of initially gyro-phase bunched distributions as they propagate away from the shock (Gurgiolo et al., 1993; Mesiane et al., 2001; Meziane et al., 2004). Often, as seen in simulations of ion reflection off the shock, the gyro-phase mixing is arrested early in the distribution's propagation upstream when the rotating gyro-phase bunched particles begin to drive large-scale MHD waves that themselves trap the distribution, locking it in phase (Thomsen, 1985; Gurgiolo et al., 1993). Phase-locked electrons have been observed well upstream of the shock in the presence of whistler waves



(Gurgiolo et al., 2005). The process is generally used to explain observations of phase-bunched distributions made at distances upstream of the shock beyond where gyro-phase mixing should have led to isotropization.

Return distributions contain sufficient free energy to drive a number of instabilities commonly
observed in the foreshock. These include observations of MHD and ULF waves (Hoppe et al., 1981; Greenstadt et al., 1995), ion and electron cyclotron waves (Smith et al., 1985; Kis et al., 2007), whistler waves (Hoppe and Russell, 1980; Zhang et al., 1998), ion acoustic waves (Gurnett and Frank, 1978), and Langmuir waves (Bale et al., 1997). Many of these are important in the preheating and breaking of the solar wind prior to its interaction with the shock.

It is often through simulations that processes thought to act in the reflection and acceleration of the solar wind come to light (e.g., gradient drift at the shock (Leroy et al., 1981; Krauss-Varban and Wu, 1989)). Direct observations of post reflected distributions are a second source (Burgess, 1987; Kucharek et al., 2004). There are few, if any, direct observations of the actual reflection process, which would be extremely useful in helping to identify the mechanisms responsible. There
are probably multiple mechanisms that are active, either singularly or in concert, in the reflection process (Fitzenreiter et al., 1996; Yuan et al., 2007; Savoini et al., 2010). One of the most commonly invoked mechanisms is reflection through magnetic mirroring (Burgess and Schwartz, 1984; Leroy and Mangeney, 1984; Burgess, 1987), which occurs as the solar wind approaches the stronger shock magnetic field. However, as we will demonstrate below, magnetic mirroring it does not appear to
play a major role in the reflection of the strahl. Mirror-reflected distributions are distinctive and can easily be identified and readily differentiated from return particles that have leaked though the shock from the magnetosheath (Larson et al., 1996).

Almost all reflected particles undergo energization in the reflection process. This occurs in the repartition of the incident particle's parallel and perpendicular velocity with respect to the magnetic
field in the reflection process. The energization is driven by changes in the perpendicular velocity that shifts the particles guiding center position with respect to the VxB electric field (Sonnerup, 1969; Paschmann et al., 1981). Under certain conditions, reflections can also act to decelerate the particle (de-energization).

In this paper we closely examine the reflection of solar wind electrons off the shock. In particular,
we are interested in what portion of the solar wind is being reflected and where the reflection occurs. We will also demonstrate a very simple and novel method for determining when a spacecraft is inside the foreshock that we developed in conjunction with this study. The method does not require knowledge of the shock location, any knowledge of the parameters associated with the shock, nor any modeling.





## 2 Data

The data used in this study were provided by a number of experiments on-board the Cluster spacecraft. These include: the **P**lasma **E**lectron **A**nd **C**urrent **E**xperiment (PEACE) (Johnstone et al., 1997; Fazakerley et al., 2010), the **F**lux**g**ate **M**agnetometer (FGM) (Balogh et al., 1997; Gloag et al., 2010), the **E**lectric **F**ield and **W**aves (EFW) experiment (Gustafsson et al., 1997; Khotyaintsev et al., 2010), and the **W**aves of **Hi**gh frequency and **S**ounder for **P**robing of **E**lectron density by **R**elaxation (WHISPER) (Décréau et al., 1997; Trotignon et al., 2010).

The primary data used in this analysis are from PEACE, which consists of two hemispherical electrostatic analyzers designated HEEA (High Energy Electrostatic Analyzer) and LEEA (Low Energy Electrostatic Analyzer). They are located $180°$ apart on the satellite. The analyzers differ only in their geometric factors (HEEA's geometric factor is the larger one). Despite their acronyms, both can cover identical energy ranges from 0.6 eV to 26 keV. The analyzers' fields of view are perpendicular to the spacecraft spin axis, which is about $5°$ off GSE-Z and cover $180°$ in elevation in 12 sectors. A full $360°$ in azimuth is covered in one rotation of the spacecraft so that a three-dimensional snapshot of the electron distribution is accumulated once per spin (∼4s).

Because of telemetry restrictions PEACE generally returns only a subset of the total data collected, even in burst-mode. Exactly what is returned depends on the instrument mode, which can be separately commanded for each analyzer on each of the four spacecraft. The telemetry rate, as well as the amount of data being returned, determines the time cadence at which full three-dimensional distributions are downloaded. During the time intervals used in this paper, all satellites were operating in burst-mode telemetry and PEACE was returning one 3D distribution per spin. Each distribution consisted of 30 energy bins with each bin divided into 6 or 12 elevations and 32 azimuths. In general, the C2 and C4 experiments returned 12 elevation bins while C1 and C3 returned only 6. Priority was given to using data from either C2 or C4 as the higher polar resolution greatly improved the registration of the strahl with respect to the magnetic field.

PEACE data are used to characterize the electron plasma using both moments and visualization tools that allow one to highlight aspects of the morphology in velocity space of the electron 3D Velocity Distribution Function (eVDF). Depending on the lower energy threshold coupled with the spacecraft potential there are times when the lower energy portion of the core electron population cannot be sampled. The FGM 5 vector per second data are used both to characterize the local magnetic field and to compute the shock normal. Both the EFW and WHISPER are used in the calculation of the electron moments; EFW provides the spacecraft potential used to correct the measured electron energy and WHISPER provides the flags necessary to filter out times during which the computed moments may be contaminated by local perturbations created by WHISPER active sounding.

All of the data used in the analysis presented in this paper were obtained from either of two open data archives: the Cluster Science Archive (CSA) (https://csa.esac.esa.int/csa-web/) and the Mullard




Space Science Laboratory (MSSL) Cluster Archive (http://www.mssl.ucl.ac.uk/missions/cluster/about_peace_data.php). The CSA provides data in either CDF or CEF format while the MSSL archive returns data in IDFS format that is suitable for use with the UDF Analysis, $\phi-\theta$ Visualization, and Moments packages
used for all the analysis presented here.

### 3    $\phi-\theta$ Plots and Moments

A majority of the supporting analysis and conclusions in this paper come from either estimates of the plasma moments of the individual electron populations or from various features observed in the $\phi-\theta$ plots (also referred to in the literature as "sky maps"). The $\phi-\theta$ plots are a good plot
format for investigating three-dimensional features in eVDFs. For a full description, see Gurgiolo et al. (2010). A detailed description of how the moments are computed as well as how to separate electron populations within an eVDF can be found in Gurgiolo and Goldstein (2016). In the analysis presented here we use a slightly modified version of the population isolation method described in the aforementioned paper. This is briefly outlined below.
Figure 1 contains 3 columns of $\phi-\theta$ plots illustrating the method used to separate the strahl and return electron populations. Only a subset of the returned energy steps are shown and each column of plots shows the same eVDF but with different masks applied. The same subset of the returned energy steps are displayed in each column. The first column has no mask applied and shows the complete $\phi-\theta$ content within each of the plotted energy ranges. The black and red traces are lines
of constant pitch-angle of $120°$ and $80°$, respectively. The second column of plots masks out all data with pitch-angles greater than $80°$ at energies $\geqq 47.9$ eV, which leaves just the return electrons. The third column masks out all pitch-angles less than $120°$ at energies $\geqq 56.7$ eV, which leaves only the strahl electrons.

The areas outside the masks have been set to zero so that the standard numerical integration
technique used to estimate the basic plasma moments can be made over each of the three columns without any modifications. This approach yields the estimated plasma parameters associated with the full, return and strahl electron populations separately. The energy integrations for the latter two populations start at 47.6 and 56.7 eV respectively. Deciding the lower energy limit at which to begin masking the data and over which the numerical integration is made is subjective and can be different
for different populations. In general, we use the energy step above the first unambiguous observation of the population and should it overlap another population (as is often the case with the strahl and core-halo) the energy at which it becomes dominant. This energy is used for an entire event unless there is a clear indication that it has shifted up or down, in which case the moment computations are terminated and restarted at the new time with the updated start energy.



## 4 Analysis Techniques, Terminology, Common Figure Formats


This section contains descriptions of the techniques used in the event analysis as well as terminology that may be unfamiliar and the plot formats used in some common figures. Presenting them separately allows them to be introduced in subsequent discussions without their having to be described multiple times.

### 4.1 Foreshock Determination


The return electron population is a common and ubiquitous feature of the foreshock. Although its source may vary from reflection at, to leakage through, the bow shock, its presence or absence essentially determines if a spacecraft in the upstream is in the foreshock or in the solar wind. As knowledge of the spacecraft location with respect to these two regions plays an important role in this study, we have developed an effective and simple proxy using the density of the return electron population to provide this information. The method is continuous, sensitive, and can easily be automated to interface with most analysis tools.


While in practice one can use the $\phi - \theta$ plots to essentially perform the same task, it is both tedious and time consuming to do so even for relatively short intervals (e.g., 10 minutes). The algorithm developed here is based on the surmise that return electrons only exist in the foreshock and not in the solar wind. Under this assumption, if a blind computation of the return electron density is made utilizing a VDF mask of the type shown in the 2nd column of Figure 1, then one expects to see a bi-modal density pattern, high density when the spacecraft is located in the foreshock and low density when it is located in the solar wind. Indeed, this is exactly what is seen. Figure 2 shows the return density computed for a 40 min period upstream of the bow shock. The left-hand panel in the figure contains the time variation of the density and the right-hand panel which illustrates the bi-modal nature of the data is a plot of the Probability Density Function (PDF) of the same data. To separate the foreshock from solar wind regions, a break point is set such that measurements taken when the return density is above the breakpoint are ascribed to the foreshock and below to the solar wind. The correctness of the break point should be verified using $\phi - \theta$ plots. For the example shown, the break point was set to $0.09\,cm^{-3}$, which was used to set the color in the left-hand plot (red when the spacecraft was in the solar wind). The density seen in the solar wind results from the encroachment of other populations into the area defined by the return mask. A double break point can also be set such that densities below the lower value signify that the spacecraft was in the solar wind and densities above the upper break point that the spacecraft was in the foreshock. The region defined by return densities between the two breakpoints can be classified as undetermined. The breakpoint(s) need to be set on an event by event basis because of differences in the average density between events. The breakpoints also need to be reset anytime the energy integration limits are changed. The determination of the spacecraft location (foreshock or solar wind) has a temporal







resolution equivalent to the cadence at which the 3D eVDFs are returned (4s in the example in Figure
2), which allows for easy identification of rapid motion of the foreshock boundary. Two final notes:
if the spacecraft is only in one region for an entire event one needs to use a $\phi-\theta$ plot at least at one
point in the event to determine if the spacecraft is in the foreshock or solar wind; and if using the
density solely as an indicator of region (as opposed to a quantitative measure of the return density)
the numerical integration can be started at any energy step that is above the lowest energy at which
a return signature is seen the $\phi-\theta$ plots, preferably higher as that tends to produce better separation
between the pseudo return densities in the solar wind and return densities in the foreshock.

### 4.2  Shock Normal

To analytically derive either the expected pitch-angle spread of the return distribution or the expected
energy gain in the reflection process requires an estimate of the shock normal. In this study this is
done using the method described in Shen et al. (2007), which is based on the assumption that the
shock normal is anti-parallel to the gradient of the magnetic field within the shock front. The gradient
is obtained using the 5 vector per second FGM magnetic field data from all four Cluster spacecraft
following the method outlined in Gurgiolo et al. (2005). The normal vector is returned in component
form in GSE coordinates with a $1\sigma$ deviation given for each component. This method is only valid
when the four Cluster spacecraft are in a good tetrahedral configuration, which we define as having
a QGM > 2.8 (Robert et al., 1998).

### 4.3  Energization Through Reflection.

To make direct comparisons between the return and strahl populations, it is necessary to remap the
return electrons in energy to account for any energy gained in the reflection process. For example
if the energization on reflection is a factor of two then the strahl density above $\epsilon$ eV should be
compared to the return density above $2\epsilon$ eV. The energization can be estimated using the methods
described in Sonnerup (1969); Paschmann et al. (1980). The method described in Paschmann et al.
(1980) is essentially a 3D generalization of that presented in Sonnerup (1969). In this work, we
make particular usage of Eq(9) in Paschmann et al. (1980) for this purpose. At the energy range
used in our analysis we tacitly assume that the reflected electrons are associated with the strahl. This
allows the incident velocity angle with respect to the normal in Eq(9) to be directly obtained from
the orientation of the local magnetic field just upstream of the shock since the strahl fluid velocity is
either parallel or anti-parallel to the magnetic field. The energization is computed for a number of
shock normals and upstream magnetic field orientations by varying both parameters within the one
sigma band about their measured component values. Note: variations in the magnetic field translate
directly to variations in the incident population flow direction.





### 4.4 Estimated Pitch-Angle Spread of the Reflected Electrons

Under the assumption that the shock is a solid reflective surface, that all reflections are specular,
and that the incident strahl guiding center velocity is parallel or anti-parallel to the average magnetic
field (depending on whether it is pointing sunward or anti-sunward), it is possible to estimate the
pitch-angle spread expected of the return electron distribution. The width is estimated by varying
both the shock normal and average magnetic field components within their $1\sigma$ bands (as done when
estimating the energization in reflection) coupled with a spread in the incident strahl velocity deter-
mined from its maximum pitch-angle spread as obtained using the $\phi-\theta$ plots (viz., black outline in
the first column of plot in 1). The estimates of the return pitch-angle spread can at times be a bit
high when compared to what is observed in the $\phi-\theta$ plots because we use a strahl pitch-angle limit
that is generally on the high side to ensure that the total distribution is included in the limits.

### 4.5 Crossover Energy

In both the solar wind and the foreshock there is often an energy below which the strahl begins
to overlap the core-halo and the two populations cannot easily be separated, which we refer to as
the crossover energy. Although rarely needed, it is possible to define a crossover energy for the
return and core-halo populations. The crossover energy (or sometimes the energy above it) defines
the starting energy used in the plasma moment integrations of the isolated populations. In the $\phi-\theta$
plots the crossover energy is characterized by a shift in the population location in the plot. Above the
crossover energy, the population is predominantly field-aligned and the strahl is by far the dominant
population, while below the cutoff energy the core-halo is dominant and the population shifts to
a more radial profile. This is present in all of the upstream eVDFs except at times when there is
sufficient separation between the two populations that no crossover energy exists (both populations
are fully separable at all energies). The crossover energy may vary from event to event and even
within an event with major changes of the magnetic field orientation. For the most part, however, the
crossover energy remains reasonably constant within a given event. At times when the interplanetary
magnetic field is basically radial, the crossover energy is not easily identifiable, and can be difficult
to identify when the field has only a small non-radial component. In most cases, however, the energy
at which the shift in dominance between the two populations occurs is unmistakable. The case seen
in Figure 1 is a situation in which the magnetic field has only minor non-radial components making
it difficult to identify the cross-over energy, which is probably at 56.7 or 48.9 eV where the strahl
begins to show a distention to lower $\theta$.

### 4.6 Common Figure Formats

We basically use three figure formats to illustrate the plasma characteristics of each of the selected
events. These consist of an event overview, a set of $\phi-\theta$ plots illustrating the characteristics of a



typical foreshock eVDF during the event, and a plot showing the densities of the total, return, and strahl populations across the time period. All figures of one type share a common format.

- The overview figure (e.g., Figure 3) consists of two panels. The upper panel is the full electron density and total fluid velocity with density plotted against the left axis and velocity against the right axis. Below this is a spectrogram from the PEACE elevation that is closest to the ecliptic overlaid with a plot of the magnetic field. All data are plotted at 4s resolution. Higher resolution magnetic field data are included in plots in the discussion section.

- The characterization of a typical eVDF from the foreshock for the event is shown in a set of three columns of $\phi - \theta$ plots (e.g., Figure 4). These form a set of 18 contiguous (in energy) $\phi - \theta$ plots, which together cover the energy range from 15.8 to 669.2 eV. White and red traces in each plot are lines of constant pitch-angle and are included basically to delineate where, if present in the plots, the strahl (white) and return electron distributions (red) are expected to be observed. These are also the same regions used to isolate the two populations in the estimation of their plasma moments (viz., Figure 1). The solid dot and triangle in each plot are the projection of the head and tail of the magnetic field, respectively. The plots have no smoothing applied to them and display the data in their native $\phi - \theta$ resolution.

- The density information figure (e.g., Figure 6) contains 2 columns each consisting of 4 rows of plots. From top to bottom these are the full, strahl, and return densities and the return to strahl density ratio across the event. The left-hand column of plots shows the time variation of the quantities while the right-hand column consists of plots of the Probability Density Function (PDF) of the quantities from just the foreshock periods. The average value of the plotted quantities for both regions are shown to the right of each PDF plot (foreshock, black text and solar wind, red text). If one or the other of the regions is not sampled in the event the average value is set to -1.0. In the lower three panels of plots in the first column the red portion(s) of the plot indicate times when the spacecraft are in the solar wind and the black portions times when the spacecraft is in the foreshock. This is determined directly from the return electron density as discussed in 4.1.

## 5 Observations

Although we have looked at multiple events as part of this study, we present the detailed results for only three. These are typical and illustrate most of the important features pertinent to this analysis.

### 5.1 Event 1: 2005-01-11

Figure 3 is the event 1 overview. All data were obtained from C2. The event begins in the magnetosheath and at about 15:55 UT the spacecraft passes through the bow shock and enters the fore-



shock. There is only a short stretch of foreshock (about 45s) before the spacecraft crosses into the solar wind and remains there for most of the rest of the event with the last 90s spent in the foreshock. This is explicitly shown in Figure 6.

Figure 4 shows a partial representation of a typical foreshock eVDF observed just after the spacecraft crosses the bow shock. Both the strahl and return electron distributions are field-aligned and

counter-streaming, the strahl is moving anti-sunward and the return electrons are moving sunward. The white and red traces in each plot are lines of constant pitch-angle of $120°$ ($\equiv 60°$) and $75°$, respectively. The region delimited by the white trace marks the strahl and the region delimited by the red trace marks the return population. Neither population need be present at any given energy.

In this event the crossover energy is located at about 56.7 eV. Above this energy the electron

distribution consists almost exclusively of strahl and return particles. Below this energy there is an obvious contribution of core/halo electrons that rapidly becomes the dominant population. There is still a return electron signature below the crossover energy extending probably to as low as 37.7 eV. The return electrons are noticeably non-gyrotropic at the lower energies particularly below the crossover energy where they appear as a partial ring, probably the result of a combination of phase

bunching in the reflection process and subsequent gyro-phase mixing and possibly phase locking. As we demonstrate below, it appears that above the crossover energy all of the return electrons originate from the strahl. At lower energies the return electron signature is more likely due to reflection of some percentage of the higher energy halo electrons together with some lower energy strahl that are not fully separable from the core-halo. In the $\phi-\theta$ plots the core/halo population (when present)

is centered near $(0°,0°)$ because it flows radially outward from the sun. (Recall that the spacecraft spin axis is not quite perpendicular to the ecliptic plane.) Halo and strahl electrons that overlap in velocity space will react identically to any external influences.

The angular spread observed in the return electrons in the $\phi-\theta$ plots appears to be consistent with specular or nearly specular reflection. Computation of the shock normal returns a vector of

$(0.780 \pm 0.033, 0.603 \pm 0.043, 0.121 \pm 0.098)$ in GSE coordinates, which when coupled with the average magnetic field just upstream of the shock in the foreshock gives a $\theta_{Bn}$ of $81°$. Assuming a maximum spread in the strahl pitch-angle of $60°$ we estimate there should be an $81 \pm 5°$ spread in the return electron pitch-angle distribution. The best fit to the return data as determined from the $\phi-\theta$ plots would appear to be a spread of about $75°$ (red trace in Figure 4).

Above the crossover energy the return distribution almost exactly matches the energy range covered by the strahl electrons extending one or two energy steps higher, to energies where there is no evidence of a strahl. At these energies the weak count-rate has the appearance of noise and were it not for the fact that it is observed exclusively within the region in phase space associated with the return distribution it would probably be labeled as such. The higher energy is in all likelihood the

manifestation of acceleration in the reflection. That acceleration would put the source of the return electrons below the crossover energy directly within the upper halo.



Using an ensemble of energizations obtained from Eq(9) in Paschmann et al. (1980) we place the average energization factor in the reflection process at 1.17. Figure 5 shows the results as a PDF plot. This value can be used to remap the return electron densities for comparison with the strahl. Under

the assumption that above the crossover energy the observed reflected population has the strahl as its source, we compared the estimated strahl density with energies $\geqq$70.5 eV with that of the return densities with energies $\geqq$87.5 eV (87.5 eV being the closest center energy being returned to 70.5 x 1.17 = 82.5 eV). We did this by computing the strahl and return densities within each returned energy band and then summing the densities over the energies $\geqq 70.5$ eV for the strahl and $\geqq 87.5$

for the return electrons. This mapping is equivalent to an energization factor of 1.24. The results are shown in Figure 6 beginning from just after the shock crossing to 16:30 UT. For this energization mapping, the ratio of return to strahl electrons is 1.04. This was unexpected as the implication is that at least above 70.5 eV there is statistically a full reflection of the strahl at the shock. As a check we increased the starting energies of the density summation up one energy bin for both the strahl

and return populations (to 87.5 eV for the strahl and to 110.1 eV for the return electrons which corresponds to an the energization of 1.26), which gives an average density ratio of 1.01.

### 5.2 Event 2: 2005-02-06

Figure 7 shows the overview of event 2. As in the previous event, the spacecraft begins in the magnetosheath and a little before 15:45UT passes through the bow shock and into the foreshock

where it stays for a little longer than 2 minutes before entering the solar wind. There are multiple excursions into and out of the foreshock at this point over the rest of the event. Overall the spacecraft spends a significantly larger percentage of time in the foreshock than it did in the previous event, which improves the statistics in the observed return-to-strahl density ratio, as is readily apparent in Figure 9.

The spacecraft were in a good tetrahedral configuration during this event and the shock normal was estimated to be $(0.692 \pm 0.025, 0.161 \pm 0.007, 0.703 \pm 0.027)$ in GSE coordinates. The average magnetic field just upstream of the shock gave a $\theta_{Bn}$ of 70°. Using a 50° pitch-angle spread for the strahl we obtained an estimate of the pitch-angle spread for the return electrons of $74 \pm 5°$. The pitch-angle spread determined directly from the $\phi - \theta$ plots in Figure 8 was 75° ( the red trace). The

white trace used to delineate the strahl is a pitch-angle of 50°.

Comparing Figure 4 with Figure 8 shows only minor differences in the eVDF morphology between the two events. The crossover energy is however, slightly lower in this event, probably 47.9 eV. The return electron signature again extends below the crossover energy down to at least 37.7 eV and is distinctly non-gyrotropic at the lower energy end.

As shown in Figure 9 the energization factor associated with this shock is estimated to be 1.43. Figure 10 shows the electron density profiles of the different populations across Figure 7 starting just after the spacecraft exits the bow shock. In the plot, the strahl density was determined beginning from



56.7 eV and the return density beginning from 87.5 eV. This mapping accounts for an energization factor of about 1.54, which is slightly larger than the analytical estimate. The ratio (1.01) implies that there is full reflection of the strahl in this event at least for this mapping. We looked at two further energy ranges with the strahl density summation beginning at energies 70.5 and 110.1 eV and return density beginning at 87.7 and 139.1 eV (both essentially having mappings equivalent to an energization factor of about 1.57). These give return-to-strahl density ratios of 1.08 and 1.05, respectively.

### 5.3 Event 3: 2008-04-15

The overview of the third event is shown in Figure 11. Again, the event begins in the magnetosheath with a bow shock crossing at about 19:15 UT. The spacecraft then enters the region upstream of the foreshock (v., Figure 13) and remains there for the rest of the event, with the exception of a short excursion into and out of the solar wind near 19:28 UT. Unfortunately, for this event the 4 spacecraft were not in a good tetrahedral configuration and no estimate of the shock normal was possible. But an estimate of the shock normal is not critical for this event as we have demonstrated in the first two events that the analytically derived values of both the angular width of the return distribution and the reflection energization factor closely match what can be obtained directly from the data.

For this event we obtained a pitch-angle spread of $70°$ for the return electrons directly from the $\phi - \theta$ plots (v., Figure 12) and estimated the reflection energization factor from plots of the density ratio constructed with varying starting integration energies of both the strahl and return populations to be on the order of 1.58 eV. Figure 13 shows the population densities and density ratio constructed with the strahl and return densities beginning at 110.1 eV and 173.1 eV, respectively. This is equivalent to a 1.57 energization factor and gives a density ratio of 0.99. Lowering the beginning energy step in the estimation of both densities to 87.5 and 139.1 eV, respectively (equivalent to a reflection energization factor of 1.59) changes the average ratio to about 1.01. Lowering the starting energy of the two populations one energy step further (equivalent to a reflection energization factor of 1.56) increases the density ratio to about 1.1.

### 5.4 Errors

There are several recognized sources of error that can affect portions of the event analysis, in particular, the comparisons of the strahl and return densities. Primary among these possible errors is the analytical estimate of the reflection energization factor. The error is purely statistical, resulting from the use of all possible combinations of magnetic field and normal orientations within the $1\sigma$ band about the actual component measurements. This value determines the remapping in energy of the return population. As can be seen from Figures 5 and 9, the error is not large, but even small errors can significantly affect the estimation of the measured return-to-strahl density ratio, which depends on the ability to remap the return density in energy. The remapping often splits the starting



energy between two energy bins. Depending on which starting point energy is selected in summing
up the density moment, one will either over or underestimate the return density, and that factors into
the average return-to-strahl density ratio. Another statistical error arises when the spacecraft spends
insufficient time in the foreshock to amass a good statistical number for the average return densities.
Such errors rise to an overall larger standard deviation in the density ratio. A final source of error
that can effect the density ratio arises when only a single estimate of the energization factor per event
is obtained at the shock crossing. Changes in the orientation of both the shock normal and upstream
magnetic field over the course of an event in reality will continuously change the energization. The
remapping used to mesh the return and strahl density estimates is unlikely to remain constant across
the event as we currently assume.

## 6   Discussion

Unlike the solar wind, the foreshock is a region that embodies what might be best described as a
continuous presence of low to medium level turbulence. This turbulence arises from back-streaming
ions and electrons created from the reflection of the incident solar wind off the shock. Both pop-
ulations are field-aligned and together they provide the necessary free energy to drive a number of
instabilities which, e.g., can generate MHD and ULF waves along with Langmuir waves. The insta-
bilities are responsible for the initial scattering and preheating of the solar wind as it approaches the
shock.

   Ion reflection off the shock is better understood than that of electrons, primarily because it has
been studied in greater detail. Simulations have played a substantial role by providing a large number
of possible reflection mechanisms, but the simulations have not provided information as to which
mechanism(s) are dominant or most important. The results of this study indicate that the reflected
electrons are primarily the strahl electrons, which may place limits on some of the available reflection
mechanisms. Like the ions, the electrons gain energy in the reflection, as can be seen at the upper
energy $\phi - \theta$ plots in Figures 4, 8, and 12. Although the counts are weak in these plots, it is obvious
that the return electrons extend a few energy bands higher than the strahl due to energy gained in the
reflection process.

The increase in angular width of the return electrons over the incident strahl is consistent with a
specular reflection. The same is true of the formation of the partial ring distributions often observed
in the lower energy $\phi - \theta$ plots (cf., the top three plots in the first column of Figure 8). This is proba-
bly the result of gyro-phase bunching in the reflection process and, if observed far enough upstream,
would imply the presence of an active phase trapping mechanism (Gurgiolo et al., 2000, 2005).
Electrons phase-mix extremely rapidly, and, coupled with their higher speed, should isotropize sig-
nificantly closer to the bow shock than do the ions.

   While the ratio of the incident and reflected population densities is highly suggestive of a full



reflection of the strahl, at least above the crossover energy, there are other indications that lead to the same conclusion and provide even more information about the process. Two of the more obvious questions that can be addressed include where does the reflection occur and how thick is the reflecting region? Answering both questions can be approached by using a series of $\phi - \theta$ plots to monitor the strahl as it approaches the bow shock and observe the changes that occur in the eVDFs.

From an idealistic observational point of view if there is a full reflection of the strahl one might expect that at some point both it and the return electron population would just drop out of the $\phi - \theta$ plots. Where this occurs would then be the reflection point. If the reflection were sufficiently rapid, occurring within one or two gyroradii, given the spacecraft velocity and the cadence of 3D eVDFs being returned, the reflection would appear from the PEACE data to be almost instantaneous. On the other hand the strahl might be observed to gradually weaken until it is no longer a significant population. The time over which that occurred would represent the thickness of the reflection region.

There are drawbacks to this approach. If there isn't a full reflection of the strahl, then some fraction of it will penetrate into the downstream region, but the return population should still vanish in the $\phi - \theta$ plots after the reflection point. The problem is that it is not obvious that it does. No matter whether the strahl is fully or partially reflected, there will always be a plasma presence downstream of the reflection point and this plasma will, regardless of its source, either be field-aligned or moving radially with the solar wind. Any other flows require the identification of additional forces such as might be provided by the cross shock potential. The problem becomes how to determine if populations seen downstream of the reflection point are the same as or different from the populations seen upstream of the reflection point, in particular, the return electrons.

As there is no guidance on how to differentiate field-aligned electrons downstream of the reflection point from the return electrons upstream of the reflection point we developed a basic set of criteria to use to accomplish this. The criteria stipulate that once it is obvious that reflection point has been crossed an observed population might be a signature of the reflected strahl if:

1. The population in question is field aligned and moving back into the upstream.

2. The population covers approximately the same energy range as the return electrons upstream of the reflection point.

3. The population has roughly the same angular spread as it the return electrons upstream of the reflection point.

4. The population is approximately continuous in time, i.e., not intermittent.

The result of this type of analysis is shown in Figure 14, which contains a column of three spectrograms from PEACE elevation zone 5 (near ecliptic view), one plot corresponding to each of the three analyzed events. Each spectrogram is overlaid by the local magnetic field. These convey similar information to that shown in Figures 3, 7, and 11 but only cover about 2 to 3 minutes of time





about the shock crossings to provide for a higher resolution view. The PEACE spectra have been spin averaged and the magnetic field has a 0.2s temporal resolution. In each plot the foot of the shock and shock ramp is clear.

Each crossing has associated with it a pair of arrows labeled **1** and **2**. The position of arrow 2 marks what can be thought of as the most forward boundary of the foreshock as it nears the shock. It is the time of the last unequivocal observation of an electron eVDF that contains both return and strahl populations. At times earlier than arrow 2, there is no observable strahl signature in the $\phi-\theta$ plots, the implication being that arrow 2 marks the location of the reflection point. The absence of the strahl before the reflection point is indicative that the strahl is fully reflected. There is, however, a return electron signature at and after the time of arrow 1. The return population is either absent or questionable at times earlier than this. It should be emphasized that the locations of these arrows are somewhat subjective (arrow 1 much more so than arrow 2).

We are reasonably certain of the placement of arrow 2 in all of the plots (it is obvious when the strahl drops out of the $\phi-\theta$ plots), the location of arrow 1 is more problematic. This arises from the identification of electrons observed prior to arrow 2. We use the 2005-02-06 event to illustrate the problem. Figure 15 contains 7 time sequential columns of $\phi-\theta$ plots each characterizing an eVDF at seven energy steps between 30.1 and 342.7 eV. The energy steps are not sequential but chosen to give the best overall view of where changes on the eVDF are occurring. The figure covers the time frame between 15:43:40.6 to 15:44:05 UT, which includes both arrows in the middle plot in Figure 14. The time of arrow 2 is covered in the seventh column and the time of arrow 1 is covered in the fifth column. The first sweep in each plot is against the left-hand axis which marks the starting time.

Column 7 in the figure is unquestionably from the foreshock as it exhibits both a return and strahl signature down to at least 70.5 eV. The crossover energy is at 56.7 eV and below this there is just a core-halo and a return population signature. The return electrons are anisotropic at the lower energies and that may extend to energies above where it is seen farther upstream (e.g., Figure 8). This perhaps is to be expected so close to the reflection point but other than this there is little difference between what is observed in column 7 from what is seen in the eVDFs that follow it in time (not shown). The eVDF in column 6 is very similar to that in column 7 but with a weaker and less defined strahl signature and could probably represent the last foreshock eVDF as well as that of column 7 indicating that the reflection occurs in a very narrow transition region.

The first 5 columns show no evidence of a strahl population. There is, however, an electron presence within the strahl mask region, but this is more of a general background than a distinct population. The eVDFs do, however, show sunward propagating field-aligned beam(s) that may or may not consist of reflected electrons. (Although, if they are reflected electrons, it is unclear what the incident particle population is.) One possibility would be that they are upper energy halo electrons. Column 5 appears to be the first column of plots that shows a return electron distribution that is consistent with what is seen after it although it is a bit weaker and less defined at the lower energies.





The first 4 eVDFs contain back-streaming electron populations but with prominent and consistent deviations from what is seen in the last three columns of plots. There is, for example, no significant core-halo presence and at times (notably in the second column), no back-streaming electrons at the lower energies. The same features can be seen in earlier eVDFs (not shown here), but they are highly intermittent and exhibit considerable variation in energy and intensity, the latter peaking as would be

expected with the increase in density at the shock. Observationally, we do not claim the presence of a return electron signature in the eVDFs as seen in the foreshock earlier than column 5. While this is subjective, it is probably not off by more than 4s. Consequently, while Figure 15 shows that arrow 2 in Figure 14 is reasonably well placed, the placement of arrow 1 at 15:43:57 UT is less certain.

Even with the uncertainty in the placement of arrow 1 in the center plot of Figure 14, it is clear

that contrary to the general descriptive phrase "reflection off the shock", the reflection is actually occurring within the foot of the shock and not at, in, or behind the shock front itself. The same is true of the remaining two shock crossings in the figure, although the reflection in event 1 is very close to the ramp. These observations effectively rule out magnetic mirroring as a source of the reflection. First, because there is insufficient $\Delta B$ in the foot of the shock to account for any

significant mirroring, and second, because there is no observed transition of any portion of the strahl into the region downstream of the reflection point. The strahl are reasonably field-aligned, albeit with a $40°$ to $60°$ spread in pitch-angle, which would preclude the mirroring of a significant percentage of the population. In each of the analyzed events, mirroring would not start until about a pitch-angle of $40° - 50°$ (loss cone), which should leave the bulk of the strahl to penetrate downstream, which is

not seen seen in the data. What, exactly, is causing the reflection at the foot of the shock is unclear. At least above the crossover energy, the reflection mechanism is very efficient.

## 7 Conclusions

One of the results of this study has been the algorithm developed to help determine when the data are in the foreshock as opposed to the solar wind. That algorithm is based on the assumption that return

electrons only exist in the foreshock. Consequently, when one sees a bi-modal density pattern, the spacecraft is in the foreshock. When the density is low and not bi-modal, then the spacecraft is in the solar wind. The pattern is shown in Figure fig:FsSWmask. Consequently, to separate the foreshock from solar wind, one sets a breakpoint such that one is in the foreshock when the density is higher than the breakpoint, but one is in the solar wind when the density is lower than the breakpoint.

Studies of the reflection of the ion solar wind have played a major role in our understanding of not only the role of reflections in the physics associated with the foreshock, but also how it occurs. However, some of what we have learned from those studies may not be applicable to electrons. The major difference between the electron and ion solar wind comes from the strahl. Unlike the ions and the electron core-halo, both of which flow radially outward from the sun, the strahl is field aligned.

Except for times when the interplanetary field is highly radial, the reflection of the strahl need not



mimic that of either the ions or of the core-halo. In this study we have found that the reflection of the electron solar wind appears to primarily consist of a reflection of the strahl, which, above the crossover energy, appears to be fully reflected not at or in the shock ramp, but rather in the foot of the shock. In the first event (top panel in Figure 14) there is a small increase in |B| near the breakpoint, but that is not the case in either of the other two events (middle and bottom panels in Figure 14). Consequently, the details of precisely how the reflection occurs is not clear, but must involve primarily the field-aligned component of the distribution. It may be that one will have to study similar events with the higher resolution afforded by data from the Magnetospheric Multiscale mission (MMS). Based on the agreement between the observed spread in the return pitch-angle with analytically produced spreads, we find that the reflection is specular. Below the breakpoint energy we cannot rule out a partial reflection of the upper energy halo electrons or of strahl electrons, which may be mixed in with them. These two populations cannot be separated in our analysis at energies below the crossover energy.

*Acknowledgements.* The authors would like to acknowledge support from NASA grant NNX15AI88G. The authors would also like to acknowledge the work and role of the Cluster Science Archive (CSA) in archiving and making available data from the Cluster mission. In addition, we also thank the WHISPER, EFW, FGM and PEACE teams for providing the data used in this study. In particular, we would like to thank the PEACE team at MSSL for their continued work in updating and improving the instrument calibration especially for those data sets taken later in the mission.



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





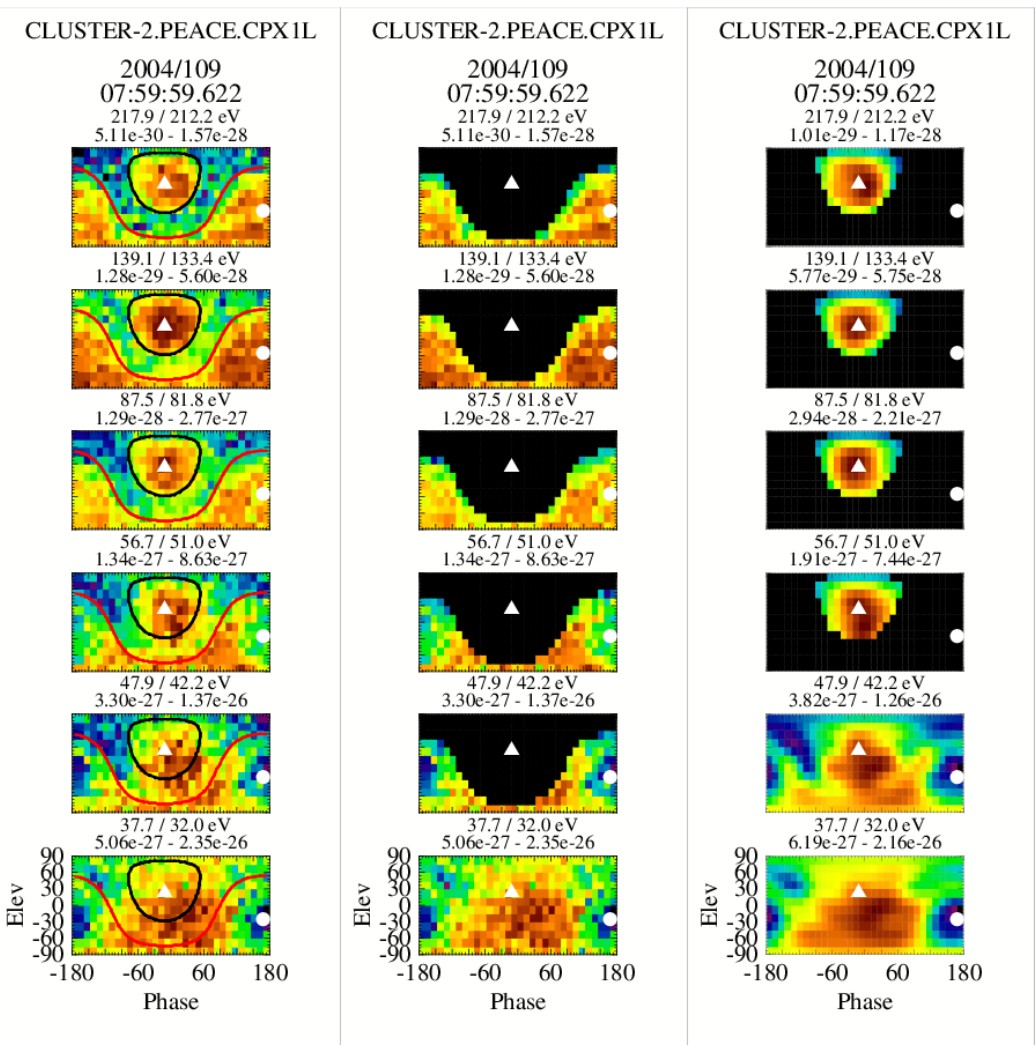

**Fig. 1.** A set of 3 columns of $\phi - \theta$ plots illustrating the use of phase space masks to isolate various electron populations. The first column shows a set of plots with no masking. The black and red traces are lines of constant pitch-angle of $120°$ and $80°$ respectively. The second column of plot masks out all pitch-angles greater than $80°$ and at energies greater than 37.7 eV leaving just the return electrons. The third column of plot masks out all pitch-angles less than $120°$ and at energies greater than 47.9 eV leaving just the strahl electrons. The solid triangle and dot in the plots are the projections of the tail and head of the magnetic field vector, respectively.





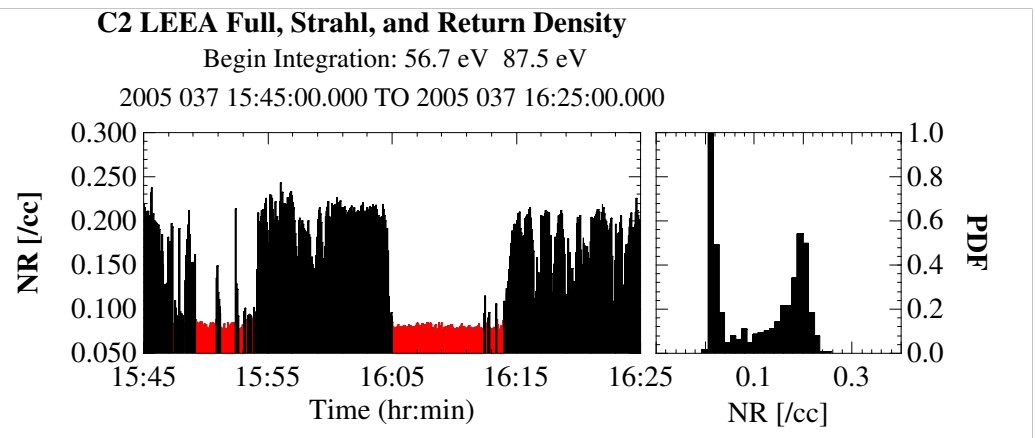

**Fig. 2.** Figure demonstrates the bi-modal aspect of the return electron density upstream of the bow shock. The right-hand plot is a PDF computed for a 40 minute stretch when Cluster-2 was upstream of the shock. The two bi-modal peaks are obvious. The left-hand plot shows the time dependence of the same data. Red indicates when the spacecraft is in the solar wind, which was determined using a 0.09 eV breakpoint in the return density.



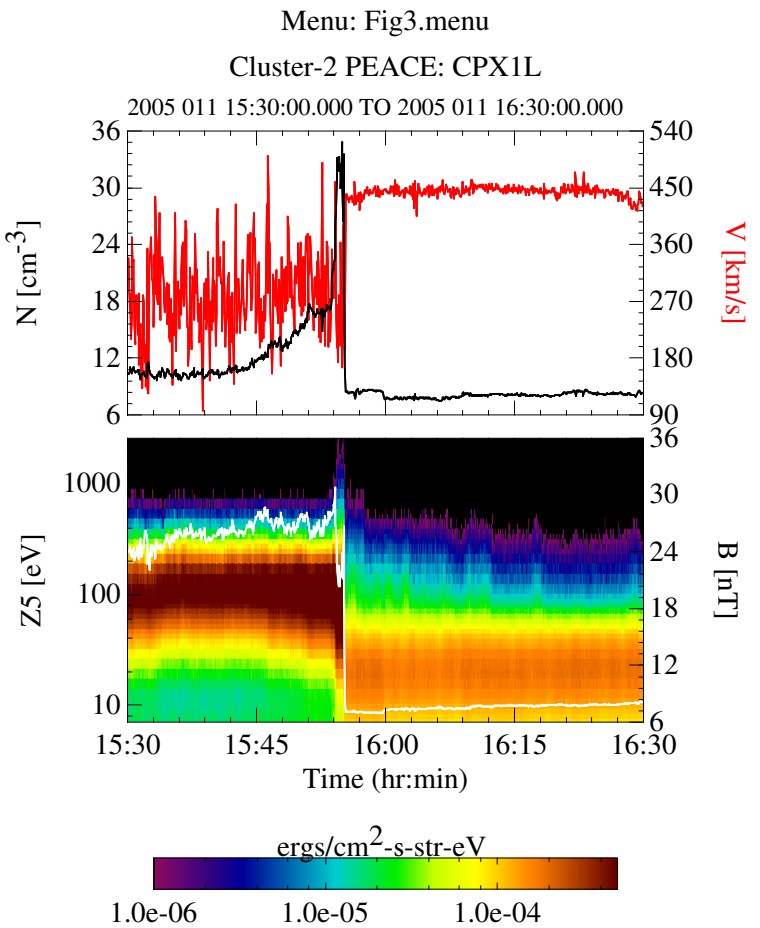

**Fig. 3.** An overview of the 2005-01-11 event. The lower panel shows an energy spectrogram of the electron data from one of the near ecliptic heads of PEACE overlaid by a trace of the magnetic field (white). The upper panel contains the full electron density (black) and the bulk fluid velocity (red). All data were from C2. At 15:55 the spacecraft exits the magnetosheath, passes through the bow shock and enters the upstream solar wind. For about the first minute the spacecraft is in the foreshock and then transitions into the solar wind, staying there for most of the remaining time period.





**Fig. 4.** A set of $\phi - \theta$ plots showing each energy step from 15.8 to 669.2 eV from a single eVDF in the foreshock. The white and red traces are lines of constant pitch-angle ($120°$ and $75°$, respectively) and are shown solely to indicate the areas where the strahl (white) and return electrons (red) might be expected to be found. The presence of either population may not exist at any given energy.





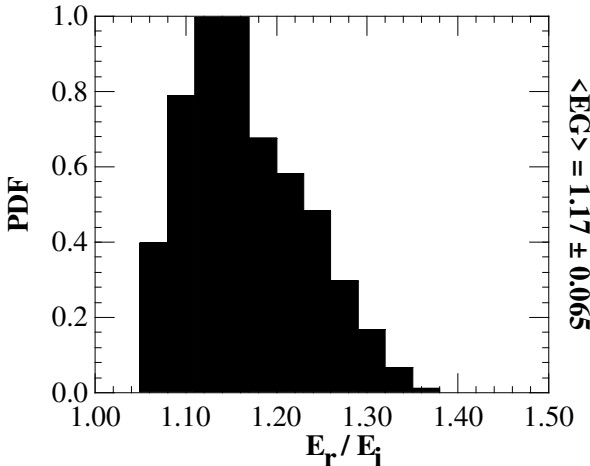

**Fig. 5.** PDF plot of the energy gain on reflection from the bow shock. The PDF was formed from the results of the energization model by varying the shock normal and magnetic field components within a $1\sigma$ band about their average values. The average energization and the $1\sigma$ value are shown to the right of the plot.



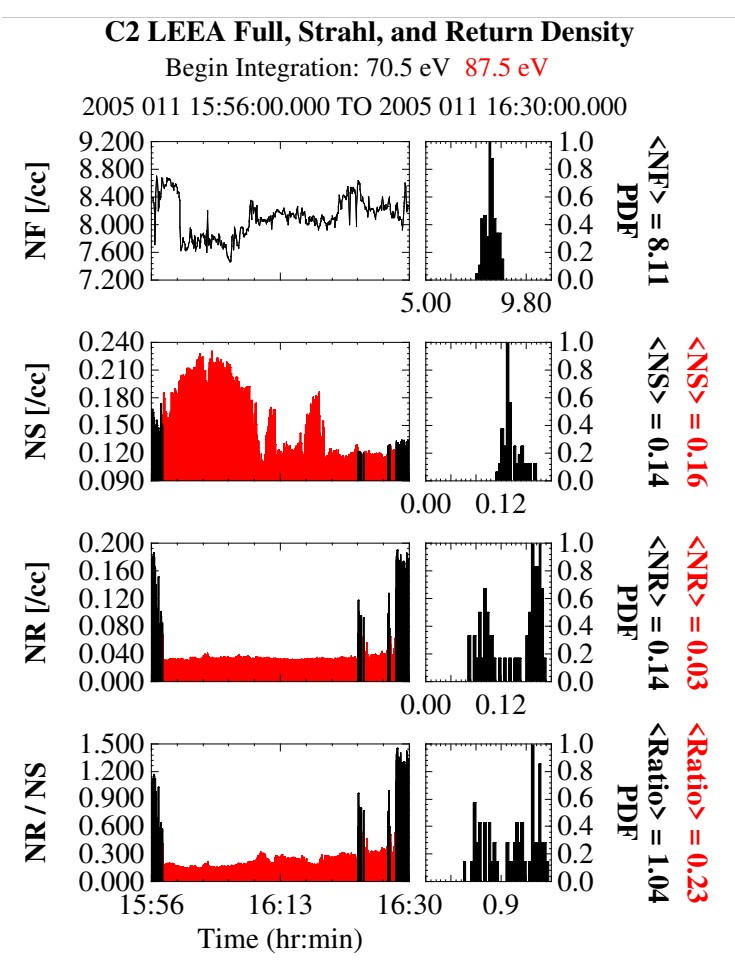

**Fig. 6.** Electron density information across the 2005-01-11 event. From top to bottom, the total, strahl and return electron densities and the ratio of the return to strahl density across the event. The left-hand column of plots show the values as a function of time. The red portions in the lower three panels show when the spacecraft was in the solar wind. The right-hand column of plots are the corresponding PDF plots of the foreshock density only. The average foreshock and solar wind densities are shown to the right of each PDF plot (red solar wind, black foreshock). The beginning energy integration used to estimate the density in each region is shown at the top.





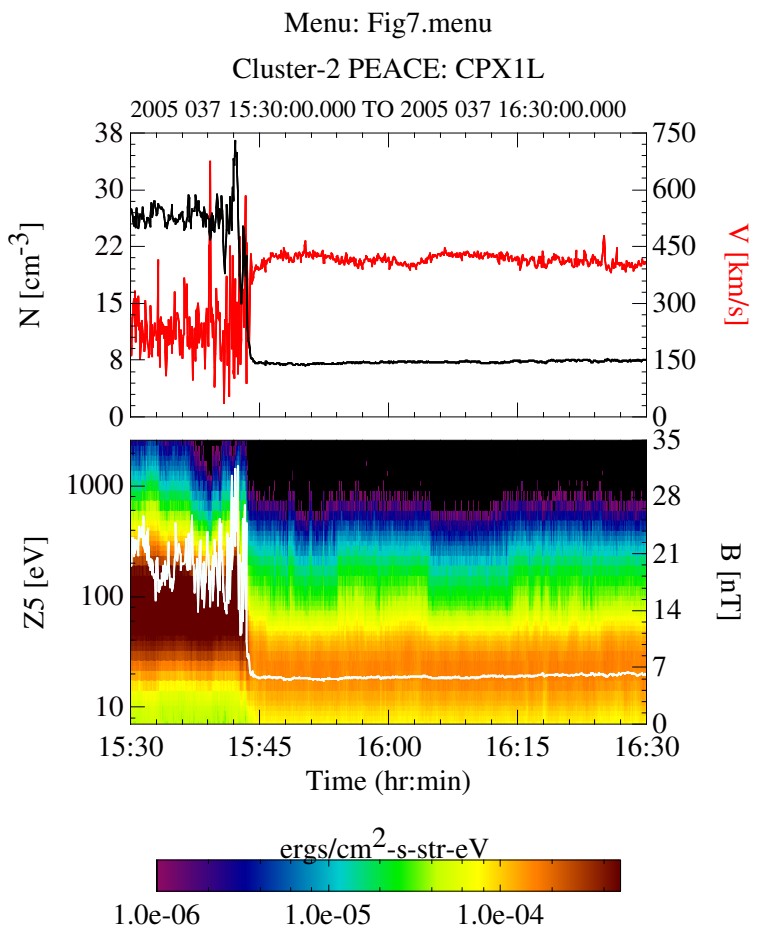

**Fig. 7.** An overview of the 2005-02-06 event. The lower panel shows an energy spectrogram of the electron data from one of the near ecliptic heads of PEACE overlaid by a trace of the magnetic field (white). The upper panel contains the full electron density (black) and the bulk fluid velocity (red). All data were from C2. At 15:45 the spacecraft exits the magnetosheath, passes through the bow shock and enters the upstream solar wind. For about the two minutes the spacecraft is in the foreshock and then transitions into the solar wind. There are two major periods of solar wind before the end of the event that can be seen in Figure 10.





**Fig. 8.** A set of $\phi - \theta$ plots showing each energy step from 15.8 to 669.2 eV from a single eVDF in the foreshock. The white and red traces are lines of constant pitch-angle (130 and 75 degrees respectively) and are shown solely to indicate the areas where the strahl (red) and return electrons (white) might be expected to be found. The presence of either population may not exist at any given energy. The solid triangle and dot in the plots are the projections of the tail and head of the magnetic field vector respectively.




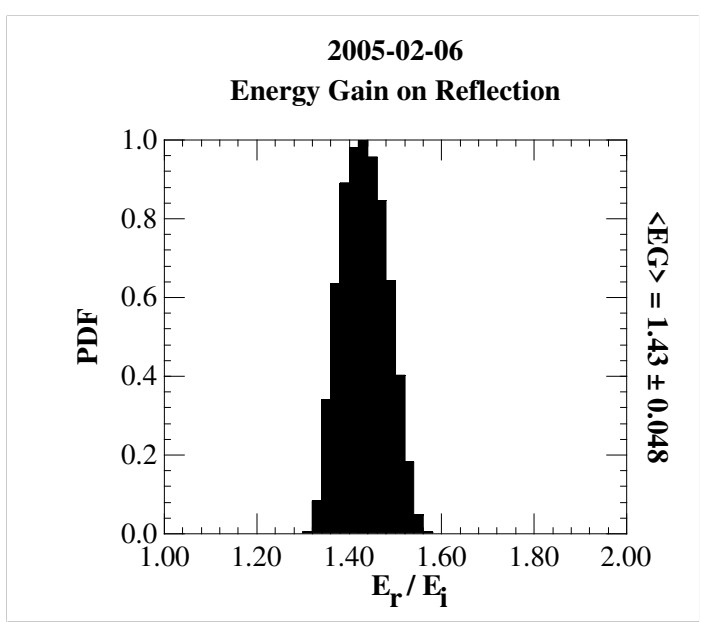

**Fig. 9.** PDF plot of the energy gain on reflection from the bow shock. The PDF was formed from the results of the energization model by varying the shock normal and magnetic field components within a $1\sigma$ band about their average values. The average energization and the $1\sigma$ value are shown to the right of the plot.





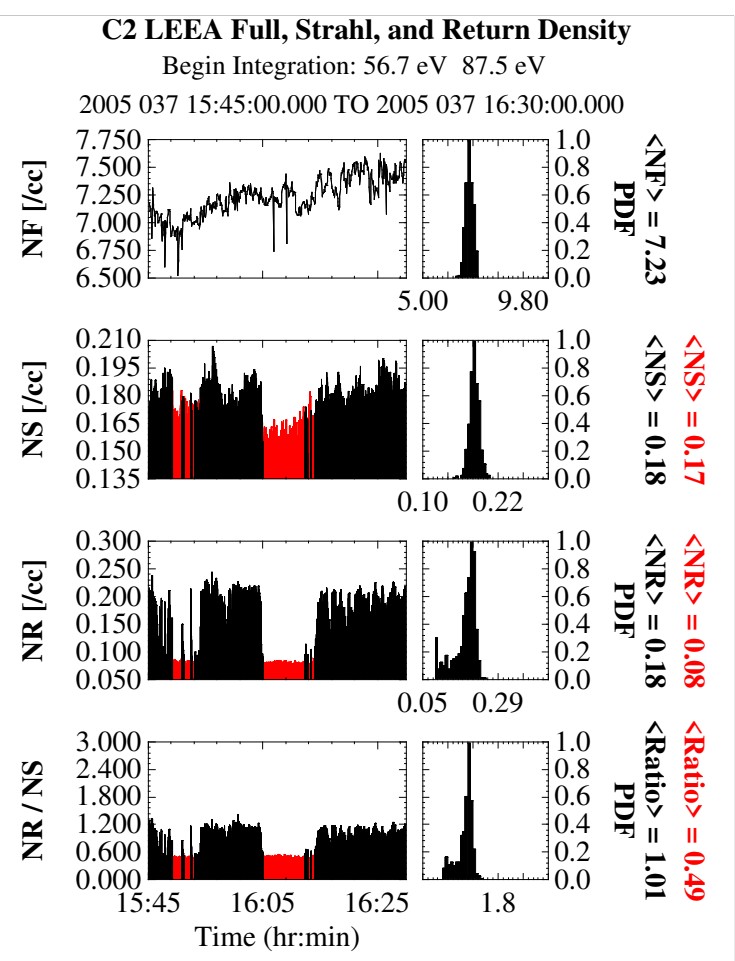

**Fig. 10.** Electron density information across the 2005-02-06 event. From top to bottom, the total, strahl and return electron densities and the ratio of the return to strahl density across the event. The left-hand column of plots show the values as a function of time. The red portions in the lower three panels show when the spacecraft was in the solar wind. The right-hand column of plots are the corresponding PDF plots of the foreshock density only. The average foreshock and solar wind densities are shown to the right of each PDF plot (red solar wind, black foreshock). The beginning energy integration used to estimate the density in each region is shown at the top.



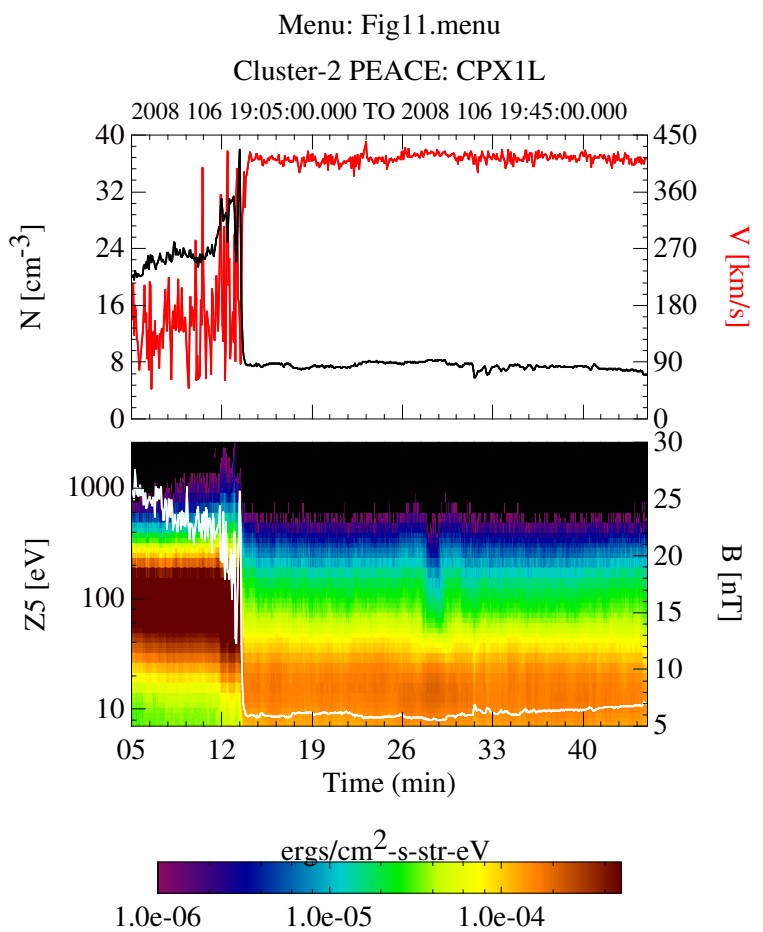

**Fig. 11.** An overview of the 2008-04-15 event. The lower panel shows an energy spectrogram of electron data from one of the near-ecliptic sensors of PEACE overlaid on a trace of the magnetic field (white). The upper panel contains the full electron density (black) and the bulk fluid velocity (red). All data were acquired from C2. At about 09:15 UT the spacecraft exits the magnetosheath, passes through the bow shock, and enters the upstream region and is in the foreshock for the remaining time period, with the exception of a brief excursion into the solar wind at about 19:28 UT.





**Fig. 12.** A set of $\phi - \theta$ plots showing each energy step from 15.8 to 669.2 eV from a single eVDF in the foreshock. The white and red traces are lines of constant pitch-angle (125 and 70 degrees respectively) and are shown solely to indicate the areas where the strahl (white) and return electrons (red) might be expected to be found. The presence of either population may not exist at any given energy. The solid triangle and dot in the plots are the projections of the tail and head of the magnetic field vector respectively.





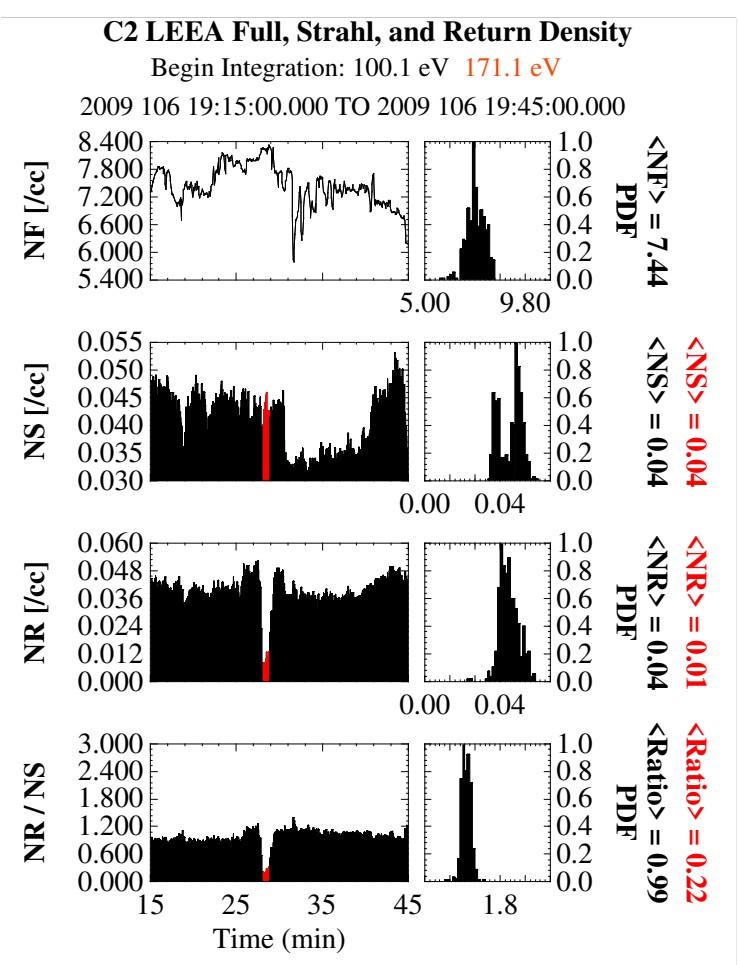

**Fig. 13.** Electron density information across the 2008-04-15 event. From top to bottom, the total, strahl and return electron densities and the ratio of the return to strahl density across the event. The left-hand column of plots show the values as a function of time. The red portions in the lower three panels show when the spacecraft was in the solar wind. The right-hand column of plots are the corresponding PDF plots of the foreshock density only. The average foreshock and solar wind densities are shown to the right of each PDF plot (red solar wind, black foreshock). The beginning energy integration used to estimate the density in each region is shown at the top.





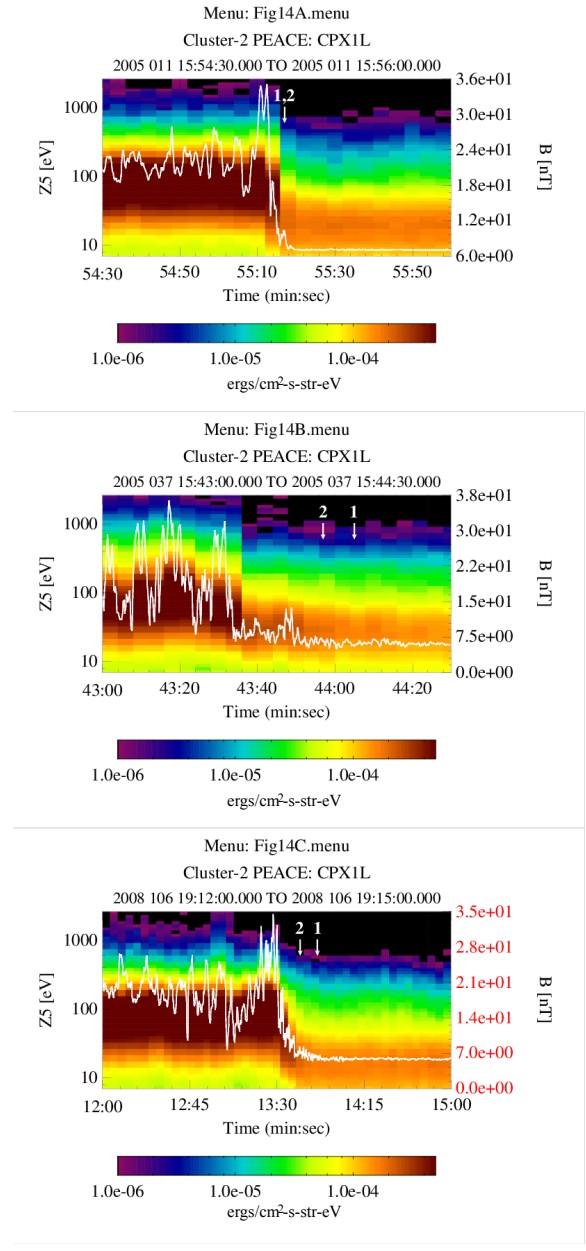

**Fig. 14.** High-resolution plots of the shock crossings for each of the three events. Each plot contains a spectrogram of the PEACE elevation Zone 5 sensor overlaid by the magnetic field. The spectrogram has a one spin resolution and the magnetic field resolution is 0.2s. There are two numbered arrows in each plot. Arrow 1 is the point at which the strahl disappears from the $\phi - \theta$ plots and arrow 2 where the return electron signature disappears. The location of the arrows is somewhat subjective, especially for arrow 2.





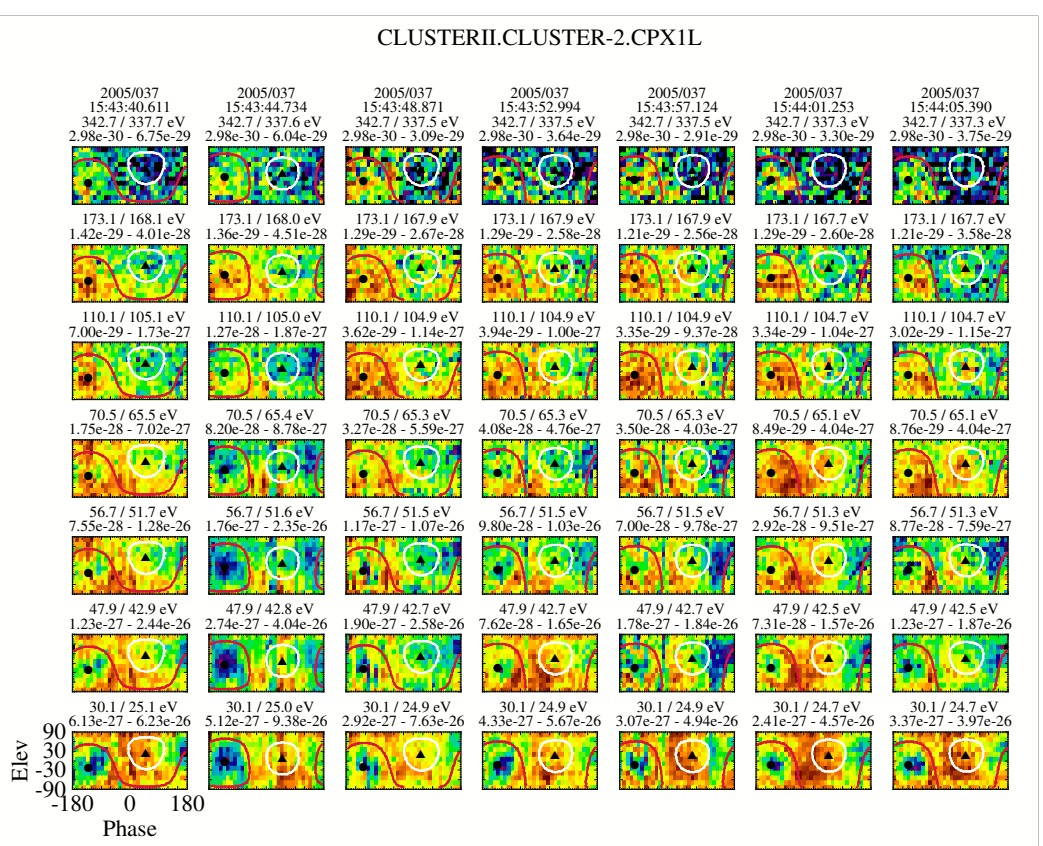

**Fig. 15.** Characteristics of 7 sequential eVDFs shown through $\phi - \theta$ plots which span the arrows shown in the center plot of Figure 14. Recall that the location of the strahl, when present, is expected within the white circle while the return electrons are expected within the red circle .