# Peer review of "Reflection of the Strahl within the Foot of the Earth's Bow Shock"

_Annales Geophysicae, 2018_

## Short Comment (SC1) · 5 Sep 2018

This comment summarise a discussion of this manuscript by the Space Plasma Physics Group at the Mullard Space Science Lab, UCL.

This paper is an interesting study on the reflection of the strahl electrons with valuable conclusions as the abstract addresses: 1. the strahl is fully reflected at the bow shock; 2. the reflection occurs in the foot of the shock.

A major thing we suggest is that this paper should be more focused on the main idea of this study. The authors could reorganize this manuscript and make it more concise for the reader. For example, in the observation section clarify which result support which conclusion point by point and make the logic flow more naturally. Perhaps moving the

observations supporting the second finding from the discussions section to the observations section could also make this main idea of this paper more focused. Another two conclusions: 3. how to determine the position of the spacecraft and 4. the reflection is specular, should also be clarified about their evidence.

The following are some minor suggestions.

In the solar wind, strahl electrons are occasionally bi-directional. A statement on this and whether it might affect these results would be helpful.

It would be helpful for the readers if the authors briefly summarise the methods from Shen et al. (2007) and Gurgiolo et al. (2005) for the shock normal determination in subsection 4.2.

In section 4.3, the assumption that reflected electrons are associated with the strahl itself uses the result of this paper. We suggest the authors present this argument more logically.

There are several subjective choices (energy ranges, reflection positions). We suggest that the authors discuss the effects of this subjectivity in the discussion section.

The authors introduce in too much detail some topics, such as: ion reflection, gyrophase bunching, foreshock waves, etc. We suggest the introduction to be more focused.

In data section, the introduction for PEACE and FGM could be more balanced (less for PEACE and more for FGM). Magnetic field data is also important for your results to come out.

Subsections of section 4 could likely be removed, such as 4.5 and 4.6.

The Figure 5 caption is not very clear, nor its description in the text. In figure 2 the title says full, strahl and return density, but the plot is only return density. In several figures, there are unnecessary text, such as figure 3,7,11, etc.

Overall, the work is interesting. We hope our feedback is helpful in the development of this paper.

---

## Referee Comment (RC1) · Anonymous Referee #1 · 25 Sep 2018

This paper presents a detailed analysis of electron data from the PEACE instrument on board the Cluster spacecraft together with magnetic field data from FGM, showing a full reflection of the field-aligned component of the solar wind electron distribution (the strahl) at the Earth's bow shock. The mechanism of reflection occurs at the shock foot, where the variations in the magnetic field are low, ruling out the possibility of mirroring. By using the electron velocity distribution functions (eVDFs) and computing the return electron densities, under the assumption that all the return electrons are in the foreshock region, authors can also determine when the spacecraft is actually immersed in the foreshock. The analysis made on the eVDFs is described accurately and is convincing, while other methods (developed in previous works) are just mentioned. I believe that the results are worth of publication in Annales Geophysicae although I

recommend to revise some parts in the description of the methodology.

1) Line 185: "correctness of the break point should be verified using \phi-\theta plots', please clarify this statement. Do you refer to the energy limits chosen for the calculation of the electron density? Since this is somewhat 'arbitrary', how can you establish that the breakpoint is the correct one?

2) Sections 4.2 and 4.3: authors just refer to other papers for the shock normal determination method and for the energization method. Since these techniques are extensively used in the analysis, I recommend to describe them with more details (for example I would suggest to report in Section 4.3 Eq.(9) in Paschmann et al., 1980). Otherwise the text results unclear.

3) Regarding the estimation of the pitch angle spread, what is the time window over which you compute the average magnetic field?

4) Line 415: please specify what you mean for "low to medium level turbulence"? Do the authors refer to an estimate of \delta B/B?

5) Figure 2: Please delete from the header 'Full, Strahl'. In the caption change '0.09 eV' with '0.09 cm^(-3)'.

6) Please discuss in Section 5.4 how all the sources of errors can influence the main results presented in the paper. For example: how can they affect the foreshock determination? I suggest to discuss more quantitatively this aspect.

7) Line 423: please add here references to simulations, as the already quoted Leroy et al., GRL 1981; Krauss-Varban and Wu, JGR 1989 Additional references: Leroy et al., JGR 1982; Scholer and Terasawa, GRL 1990.

---

## Referee Comment (RC2) · Anonymous Referee #2 · 5 Oct 2018

The manuscript presents a rather detailed observational study of the terrestrial electron foreshock using full 3D electron velocity distribution functions (eVDF) acquired on board the suit of Cluster spacecraft. Based on three specific events the authors analyze the properties of reflected electrons and further investigate where the reflection may take place in the vicinity the bow shock boundary. By use of so-called phi-theta (PT) plots, showing the full 4PI maps of the eVDF at specific energy bins, the authors conclude that (i) the majority of the reflected electron population originate from the field aligned strahl electron population, and (ii) the reflection process take place already at the foot of the bow shock ramp. In general the topic and the presented results are new and of considerable scientific interest. The manuscript as is well corresponds to international standards and is worth of publication in the journal. However, for the fi-

nal acceptance of the manuscript, I would strongly suggest to the authors to take the following comments into account.

1/ (MAJOR) While the discussion and reasoning about the "total" strahl reflection seems to be more solid and well justified by the observations and the performed analysis, the conclusions about the location of reflection are rather too strong and should be better presented as a possible hypothesis which proof will definitely require some additional analysis (like what is the justification of showing the eVDF evolution along the trajectory for event 2005-037 instead of the two other where the profile of the background magnetic field seems to be less "turbulent" and a bit more representative for a classical bow shock?; also does the missing strahl implies it was already fully reflected and not only scattered to other pitch angles?; why the reflected particles are observed even in case the strahl does not reach this location?). Therefore I strongly suggest to rewrite the conclusions accordingly and namely change the title of the manuscript.

2/ Although the "total" reflection seems to be observed the discussion should be extended like for the possible contribution of sunward propagating electrons coming from the down-stream region and accelerated eg in the bow shock. But still the observed NR/NS ratio almost equal to unity is impressive.

3/ A substantial part of the whole analysis is based on many "subjective" threshold values (limit pitch angles, density rations, integration energy intervals) which are not well explicitly defined (eg, line 157 "the energy at which it becomes dominant" does not defines what is already dominant...). Although this fact tends to be covered in the overall discussion it is completely missing in the conclusions where it should be even emphasized!

4/ For the complete picture it could be of interest to plot not only the variation of the B-field magnitude (Figure 14) but also of the individual components to see how the magnetic background is stable or not. This is highly relevant namely when discussing the B angle to the shock normal.

5/ The PT plots should display some color bar to give information about the scale of the color maps. In the present form it is impossible to see what is the level of variation and how the strahl/reflected electrons are significant wrt to the core-halo part. The authors could also consider to plot the eVDF PT plots in the solar wind frame (from the text it seems this is not the case now) which can very likely make the visual separation of the strahl/reflected particles more easy at lower energies.

6/ For a probability distribution function (PDF) I would expect Sum(PDF)=1, here this is not the case as all the displayed PDF show already the maximum value for the most probable bin equal to 1... Please correct or define what is your PDF.

7/ When plotting the NR/NS ratio, consider adding a line y=1 so the reader can better see what is the variation around the "total" reflection. Also the y-scale on Figure 13 for this ratio can be adjusted accordingly, here the max value is too high.

line 65 - It is often... a verb is missing?

line 74 - remove "it" after mirroring

line 76 - though -> through

line 129 - What is UDF Analysis, it is generally known or should be described here (or removed)?

line 178 - (also related to comment 3/) high/low density wrt what? What about to consider normalizing the NR by NT? Would it make the foreshock determination more robust?

line 212 - What is QGM? Either remove or explain a bit.

line 303 - The meaning of the last sentence is not clear even from the context.

line 308-309 - Would the return population "become more gyrotropic" (full ring) when plotting the PT plot is solar wind frame?

[Figure]

line 387 - remove "eV", the energization factor has no units

line 408 - effect -> affect ?

line 526 - are -> is

line 533-539 - This paragraph should be placed later in the Conclusions. Fisrt one should recall the MAIN results.
* * *

---

## Author Comment (AC1) · 2 Jan 2019

angeo-2018-90 reply to referee1
In this revised version, we have tried to respond to the referee's constructive suggestions.

angeo-2018-90-RC1:

*Line 185: "correctness of the break point"*
The analysis is insensitive to the precise value for the breakpoint and for the events we found, the populations are quite distinct so any additional precision would add anything significant. Also, it is perhaps worth pointing out that the events shown in this paper are the only ones that we were able to find—foreshock events in burst mode data are not very numerous.

*Sections 4.2 and 4.3: authors just refer to other papers for the shock normal determination method and for the energization method. Since these techniques are extensively used in the analysis, I recommend to describe them with more detail.*

Reviewing the method described in Paschmann 1980 would require a lengthy and complicated insertion that would merely reproduce elements of the 1980 paper. The technique is not particularly new, so we feel that the interested reader can glean the necessary details directly from the referenced paper.

*Regarding the estimation of the pitch angle spread, what is the time window over which you compute the average magnetic field?*
4 sec, i.e., one spin.

*The text around line 415 has been rewritten.*

*Figure 2: Please delete from the header 'Full, Strahl'. In the caption change '0.09 eV' with '0.09 cm^(-3)'.*
Fixed

*Please discuss in Section 5.4 how all the sources of errors can influence the main.*
The errors in the analysis are primarily statistical and the conclusions will not be substantially affected by those errors. More events would, of course, help, but there are no more events in the Cluster data. Perhaps future analyses using data from MMS will provide more insights.

*Line 423: please add here references to simulations, as the already quoted.*
Done

---

## Author Comment (AC2) · 2 Jan 2019

angeo-2018-90 reply to referees2
In this revised version, we have tried to respond to the referee's constructive suggestions.
*Angeo-2018-90-RC2*

*In this revised version we have tried to address the referee's concerns:*

*1/ … the performed analysis, the conclusions about the location of reflection are rather too strong and should be better presented as a possible hypothesis which proof will definitely require some additional analysis …*

We have changed the title of the paper and modified the discussion and conclusions. It is perhaps worth reiterating that this analysis was carried out on all the available burst mode data. To expand the analysis in the future, one would have to use data from, for example, the MMS mission. Whether a useful population of foreshock events can be found in MMS data is a subject for future study.

*3/ A substantial part of the whole analysis is based on many "subjective" threshold values…*
We have tried to be a quantitative as possible given the observational constraints. We feel that the examples are compelling. Regardless of the fine points, it is nonetheless clear that from these examples and from the phi-theta plots that the reflection of the strahl is not coincident with any increase in the magnitude of the magnetic field and, consequently, there is no evidence that magnetic mirroring is playing an important role in the reflection. To be more quantitative would require more examples, which are, unfortunately, not available.

*4/ For the complete picture it could be of interest to plot not only the variation of the B-field magnitude (Figure 14) but also of the individual components to see how the magnetic background is stable or not. This is highly relevant namely when discussing the B angle to the shock normal.*

We attach a plot of the components, but since the plot does not add any information other than confirming that the computed shock normal is a good estimation of the geometry of the event, we have chosen not to include it in the paper. Should the referee disagree, we would be happy to include it, but, frankly, don't think that it would add anything significant to the analysis.

In this plot, the data have been rotated into the shock normal coordinate system and the B_dot_n component is continuous across the shock in the rotated system (rotated Bx), indicating a relative good normal calculation.

[Figure]

*5/ (and /6) The PT plots should display some color bar to give information about the scale of the color maps. In the present form it is impossible to see what is the level of variation and how the strahl/reflected electrons are significant wrt to the core-halo part....*

The color bars are normalized for each energy. The variation with energy is too large to use a uniform color bar for each event. Consequently, in this usage, the "PDF" does not sum to unity.

*7/ When plotting the NR/NS ratio, consider adding a line y=1 so the reader can better see what is the variation around the "total" reflection. Also the y-scale on Figure 13 for this ratio can be adjusted accordingly, here the max value is too high*

Unfortunately, these plot were made using software created by the (deceased) first author and we do not have access to the scripts that generated the plots. Consequently, although we agree that it would be useful to make the changes suggested, we are unable to do so.

*line 65 - It is often... a verb is missing?*

Fixed.

*line 74 - remove "it" after mirroring*

Fixed.

*line 76 - though -> through*

Corrected.

*line 129 - What is UDF Analysis, it is generally known or should be describd here (or removed)*

That analysis package has been used in previous publications; we have removed the reference.

*line 178 - (also related to comment 3/) high/low density wrt what? What about to consider normalizing the NR by NT? Would it make the foreshock determination more robust?*

We think that the description is clear and, for the reasons mentioned above, cannot easily change the normalization, nor do we think that doing so would substantially change the determination of the foreshock.

*line 212 - What is QGM? Either remove or explain a bit.*

We've removed QGM.

*line 303 - The meaning of the last sentence is not clear even from the context.*

We don't know what is unclear here.

*line 308-309 - Would the return population "become more gyrotropic" (full ring) when plotting the PT plot in the solar wind frame?*

Since the electron speeds are much greater than Vsw, changing the reference frame of the plot (again, not easily doable), should not make any significant difference.

*line 387 - remove "eV", the energization factor has no units*

Fixed.

*line 408 - effect -> affect?*

Fixed

*line 526 - are -> is*

Fixed.

*line 533-539 - This paragraph should be placed later in the Conclusions. First one should recall the MAIN results.*

We have done that and slightly reworded the conclusions.